# Radiation as an In Situ Auto-Vaccination: Current Perspectives and Challenges

**DOI:** 10.3390/vaccines7030100

**Published:** 2019-08-26

**Authors:** Taichiro Goto

**Affiliations:** Lung Cancer and Respiratory Disease Center, Yamanashi Central Hospital, Yamanashi 400-8506, Japan; taichiro@1997.jukuin.keio.ac.jp; Tel.: +81-55-253-7111

**Keywords:** radiotherapy, cancer treatment, immunotherapy, immune checkpoint blockade, abscopal effects

## Abstract

Radiotherapy is generally considered to be a local treatment, but there have been reports of rare cases demonstrating abscopal effects in which antitumor effects have been observed in cancer lesions other than the irradiated site. This result is more likely to occur when immune checkpoint inhibitors are used in addition to radiotherapy. Certain radiation-induced chemokines and cytokines have immune-enhancing effects. Immune checkpoint inhibitors may strengthen these effects by stimulating antigen-presenting cells and effector cytotoxic T cells. To date, there is no consensus regarding the applicability of the abscopal effect in the clinical setting, including optimal methods for combining immune checkpoint inhibitors and irradiation. In this review, we highlight the evidence for interactions between cancer immunotherapy and radiotherapy and discuss the potential of such interactions for use in designing novel combination therapies.

## 1. Introduction

The systemic effects of localized irradiation are widely recognized, with generalized fatigue, anorexia and weight loss being typically observed in clinical practice. A comparatively rare clinical response to radiotherapy is tumor regression at sites distant from the irradiated region and is commonly known as the “abscopal effect” (derived from Latin for “ab” meaning “position away from” and “scopos” meaning “mark or target”), which was first suggested by Mole in 1953 [1]. Approximately 65 years later, the mechanisms underlying this effect are still not understood. Importantly, the abscopal effect seldom occurs after radiotherapy alone, with recent reports suggesting it occurs more frequently in patients being treated with a combination of radiotherapy and immunotherapy. This suggests that radiotherapy alone does not trigger a sufficiently effective antitumor immune response in patients with cancer to account for the abscopal effect. In this review, we delineate the biology and rationale for using a combination of radiotherapy and immunotherapy, and discuss future directions in utilizing this combination in clinical practice.

## 2. Radiotherapy and Immune Checkpoint Inhibitors

Radiotherapy is one of the most important modalities for treating cancer. In addition to systemic chemotherapy and immunotherapy, radiotherapy has traditionally been used for local treatment of cancer, similar to surgical treatment. Recent technological advances, including intensity-modulated radiation therapy, image-guided radiotherapy, stereotactic ablative body radiotherapy (SABR), proton therapy, and carbon-ion radiotherapy, have resulted in an improved antitumor response and reduced treatment toxicity [2,3,4,5]. However, this historical pattern in radiotherapy has recently begun to change owing to the recognition of the abscopal effect. This recognition developed along with the development and more common usage of immune checkpoint-blockade (ICB) agents. Together, these events have conferred radiotherapy with the potential to serve as a type of systemic therapy [2].

Cytotoxic T-lymphocyte-associated protein 4 (CTLA-4), also known as cluster of differentiation 152 (CD152), is a transmembrane receptor serving as an immune checkpoint and suppresses the immune response. CTLA-4 is expressed on the surface of T cells activated by dendritic cells presenting tumor antigens and inhibits the activation of T cells by binding more strongly to CD80/86 than CD28, thus inactivating T cells (Figure 1). Anti-CTLA-4 antibodies of immune checkpoint inhibitors bind to CTLA-4 to inhibit the interaction between CTLA-4 and CD80/86; consequently, T cell activity is enhanced and prolonged (Figure 1). Thus, activated antitumor T cells are induced. Blockade of CTLA-4 in a series of preclinical and clinical trials has been demonstrated to generate an antitumor immunological effect, particularly in patients with malignant melanoma. Two clinical trials using an anti-CTLA-4 agent (ipilimumab) demonstrated improved overall survival of patients with malignant melanoma [6,7]. Antitumor effects from the blockade of CTLA-4 have also been observed in patients with ovarian cancer, prostate cancer, and renal cell carcinoma [8,9,10,11].

PD-1 is a cell-surface receptor belonging to the immunoglobulin superfamily of proteins and is expressed on T, B, and natural killer (NK) cells [12]. Its structure and location were first reported in 1992 [12], and its ligands were subsequently identified [13]. Antitumor T cells infiltrating tumor tissues release cytokines, which trigger cancer cells to express programmed death ligand 1 (PD-L1). This ligand binds to the PD-1 receptor on T cells to suppress antitumor T cells (Figure 1). Thus, PD-1 was shown to play a role in aiding the escape of tumor cells from the immune response [14,15,16,17]. Anti-PD-1 antibodies of immune checkpoint inhibitors bind to PD-1 to inhibit the interaction between PD-1 and PD-L1; consequently, T cells are activated (Figure 1). The blockade of the PD-1 response has been demonstrated to have an antitumor effect in metastatic melanoma and other tumor types [18,19]. At present, anti-PD-1 antibodies have been approved as first-line therapeutic agents for advanced non-small cell lung cancer (NSCLC), chronic Hodgkin’s lymphoma, head and neck squamous cell carcinoma, gastric cancer, urothelial cancer, cervical cancer, renal cell carcinoma, and hepatocellular carcinoma [17,20].

## 3. The Abscopal Effect 

Radiation therapy, particularly SABR, can exhibit an immune-mediated effect at distant cancer sites via the abscopal effect; therefore, radiation therapy can be potentially used with immunotherapy synergistically. The response of tumors distant from the radiation site has been actively investigated, especially since the development of SABR [21,22,23,24]. Preclinical studies have demonstrated that high-dose radiation can increase T-cell priming, with CD8+-mediated impact on distant disease as well as the locally treated sites [25]. In addition, ablative doses of radiation have been found to cause an increase in type I interferon (IFN) and antitumor effects [26]. These preclinical data have generated an interest in combining SABR with various types of immunotherapy.

Tumor suppression due to the abscopal effect is probably regulated by a systemic antitumor effect caused by the release of cytokines into the bloodstream [27]. For example, Khan et al. reported that when irradiation was delivered to a region of the lungs in rats, genomic damage was observed in non-irradiated regions of the lung [28]. However, pre-treatment with Cu-Zn SOD or NO inhibitor attenuated such damaging effects in non-targeted areas [28]. Moreover, cytokines including interleukin (IL)-6, IL-1α, and TNF-α were significantly upregulated after irradiation, thus causing macrophage activation [29]. Together, their results suggest the involvement of cytokines, ROS, and NO in the activation of abscopal effects [27]. Furthermore, Camphausen et al. examined the roles of *p53* in mediating abscopal effects in mice [30]. In this animal model, both wild-type *p53* mice (C57BL/6) and *p53*-null mice received irradiation on their legs, and both Lewis lung carcinoma (LLC) and fibrosarcoma (T241) were induced at a distant location from the irradiation sites. The study reported that leg irradiation markedly reduced the growth rate of both LLC and T241 tumors in C57BL/6 mice in comparison with non-irradiated mice. However, tumor growth was not affected by leg irradiation in *p53*-null mice, indicating that *p53* is a potentially essential mediator in eliciting such effects [30]. Strigari et al. reported the *p53* status as a key predictor in the abscopal effect induced by radiotherapy [31]. In that study, wild-type (wt)-*p53* or *p53*-null HCT116 human colon cancer cells were xenografted into both flanks of athymic nude mice. A dose of 10 or 20 Gy (IR groups) radiation was delivered to a tumor induced in one side flank, leaving the other side flank non-irradiated (NIR groups). All directly irradiated tumors exerted dose-dependent, delayed, and reduced regrowth, independent of the *p53* status. Moreover, a significant effect on tumor-growth inhibition was also exhibited in NIR wt-*p53* tumors, while no significant inhibition was observed in the NIR *p53*-null tumors [31]. Hence, the abscopal effect is not probably achieved in cancer cells with *p53* loss-of-function mutations. Since *p53* mutations are predominant driver mutations in numerous carcinomas, such as lung carcinoma, breast carcinoma, brain neoplasm, colorectal carcinoma, esophageal carcinoma, and ovarian carcinoma [32,33], screening of *p53* mutations as a key predictive factor for the abscopal effect may be important in actual clinical practice. 

Several case reports published in the 1970s described the abscopal effect in patients who received radiotherapy for malignant melanoma, renal cell carcinoma, lymphoma and other tumor types [2,34,35]. Subsequently, the abscopal effect was reported to be a rare phenomenon associated with radiotherapy in certain other cancers, including breast cancer and hepatocellular carcinoma [2,36,37,38,39]. In 2016, a review by Abuodeh et al. considered 46 clinical cases of the abscopal effect associated with radiotherapy alone, reported from 1969 to 2014 [11,40]. Since the 1970s, studies have suggested a relationship between the abscopal effect and the immune system, an association that has now become well established. For example, ionizing radiation induces tumor cell death by means of immune-mediated components that affect both the immune system and radiosensitivity [2,36]. Moreover, immunotherapy has been proposed to influence the relative intensity of the abscopal effect during radiotherapy [22,25,30,41,42,43,44].

Studies conducted during the past decade have reported the abscopal effect using a combination of ICB and radiotherapy. Golden et al. reported the complete remission of NSCLC with multiple metastases to the liver, lung, bone, and lymph nodes [24]. In this case, the tumor was refractory to chemotherapy; the treatment, therefore, included radiotherapy to the metastatic lesions in the liver along with anti-CTLA-4 administration. Eventually, the multiple lesions exhibited complete regression [24]. Notably, in this case, the use of either radiotherapy or anti-CTLA-4 alone did not result in any antitumor effect [24]. In 2015, Golden et al. reported the results of a large clinical trial in which patients with metastatic solid tumors first received X-ray radiation (35 Gy/10 fractions) at one metastatic lesion and were then administrated granulocyte-macrophage colony-stimulating factor (125 µg/m^2^). This regimen was then repeated for a second metastatic lesion [39,45]. The abscopal effect was noted in 11 of the 41 enrolled patients; in the lesion showing the highest effect, the maximum tumor diameter decreased by approximately 30% [39]. Moreover, the abscopal effect was reported in another clinical trial using ICB agents. In the secondary analysis of the KEYNOTE-001 trial (NCT01295827), patients with NSCLC were administered the anti-PD-1 antibody pembrolizumab [46,47]. The patients who received radiotherapy before pembrolizumab administration demonstrated better overall and progression-free survival than those who did not. This suggested that the immunotherapy achieved improved efficacy in combination with radiotherapy [46,47]. ICB-related abscopal effects have now been described in many types of tumors, including breast, colon, lung, head and neck cancer, melanoma, NSCLC, and fibrosarcoma as well as thymic and pancreatic cancer [39,45,48,49].

## 4. Modulation of The Antitumor Effect of Radiation

Ionizing radiation damages DNA in the target cell, causing strand breaks, DNA-DNA crosslinks, DNA-protein crosslinks, and modification of the deoxyribose rings and bases. These types of DNA damage result in cell death [50,51]. However, only one-third of the DNA damage is estimated to occur due to a direct effect of the radiation. The remaining two-thirds of the damage is due to the indirect effects mediated by reactive oxygen and nitrogen species generation [45,52]. Localized radiation induces not only mechanical damage to the DNA structure, but also the release of cytokines and chemokines that leads to an inflammatory reaction and modifies the tumor stromal microenvironment. These are produced by the irradiated tumor cells, fibroblasts, myeloid cells, macrophages and can lead to various effects. For example, the induction of interleukin (IL)-6, IL-10, and CSF-1 contributes to the proliferation and invasion of tumor cells [11,53,54,55,56], whereas the secretion of pro-inflammatory IL-1β enhances the antitumor immune response [29,57]. In addition, cGAS, cyclic GMP-AMP (cGAMP), and other molecules have been reported to play certain roles in modulating the immune response [11].

The double-stranded DNA dispersed into the cytoplasm of irradiated cells activates cGAS, an enzyme that synthesizes cGAMP. This molecule activates the protein called stimulator of interferon genes (STING) and induces the production of type I IFN, both of which are essential for the activation and function of dendritic cells (DCs) and T cells [58,59]. Furthermore, these molecules also promote the release of IFN-γ (type II IFN) and upregulate VCAM-1 and MHC-I expression on tumor cells and antigen-presenting cells (APCs), which in turn promote the presentation of tumor antigens [11,26,60,61]. The radiation-induced release of inflammatory cytokines and chemokines also increases the infiltration of various leukocytes into tumor tissues, including DCs, effector T cells and NK cells, all of which enhance antitumor immune responses. Immunosuppressive cells such as regulatory T cells and CD11b+ cells, including myeloid-derived suppressor cells and tumor-associated macrophages, are also induced [11,62,63,64,65,66,67,68].

Various cytokines are induced by radiation [69]. Of these, transforming growth factor beta (TGFβ), an irradiation-related molecule, is a critical cytokine that triggers an immune-suppressive microenvironment by reducing the cytotoxicity of CD8+ T cells, suppressing CD4+ T-cell differentiation, promoting regulatory T-cell transformation, and inhibiting NK-cell proliferation [47,70,71,72,73]. The induction of interleukin-6 (IL-6), IL-10 and colony stimulating factor 1 (CSF-1) contributes to the proliferation and invasion of tumor cells and thereby displays a pro-tumorigenic role [53,54,55,56]. The production of CXC-motif chemokine ligand 12 (CXCL12) results in chemotaxis of pro-tumorigenic CD11b+ myeloid-derived cells [74], whereas the upregulation of CXCL9, CXCL10 and CXCL16 attracts anti-tumor effector T cells [75,76,77]. Macrophages are another type of leukocyte that can infiltrate the tumor microenvironment. They have two phenotypes, M1 and M2, that have different functions [78]. The classical activation of M1 macrophages induces the release of pro-inflammatory cytokines such as IL-12 and tumor necrosis factor (TNF), and thus plays a role in the killing of tumor cells. In contrast, M2 macrophages act as anti-immunogenic cells that express anti-inflammatory cytokines such as IL-10 and TGFβ, which subsequently inhibit the function of effector T cells and hence favor tumor progression [79].

The effect of radiotherapy on the tumor microenvironment is extremely complex, even exerting opposing effects on the host immune system (Figure 2). Radiotherapy may, in part, induce cancer cells to secrete pro-inflammatory cytokines, which then recruit T cells to achieve systematic anti-tumor effects [80,81]. However, this is not highly probable because regulation of the abscopal effects depends on a delicate balance between immune suppression and immune activation [80]. Unfortunately, radiotherapy alone is seldom successful in shifting the balance towards immune activation, as reported through rare cases of abscopal effects in the clinical setting [82]. However, with immune boosting therapy, abscopal effects can be markedly improved, as observed among one-third of patients in previous studies [82]. Therefore, the immune environment of specific tumors such as the availability of local dendritic cells and patient immunity are potentially important determinants of abscopal effects [82]. Various types of cytokines and chemokines play different roles in modulating the immune response, either pro-or anti-immunogenic, and maintain a net balance in the tumor milieu, much like occurs in the normal body. Importantly, under certain conditions, radiotherapy can reprogram the anti-immunologic tumor microenvironment, making it more conducive for APCs and T cells to be recruited and function, thereby inducing tumor cells to be recognized and eradicated more easily by the immune system. Furthermore, levels of neoantigens may influence abscopal effects. A high tumor mutational burden (TMB) increases neoantigen levels (Figure 3) [83,84]. Neoantigens released from dying tumor cells increase tumor immunogenicity, which is suggested to prime effector immune cells in the tumor microenvironment [85]. Thus, cancers with higher TMB, rather than those with a lower TMB, release more neoantigens in response to irradiation that potentially intensify abscopal effects [86]. Ongoing research aims at identifying strategies for tipping this balance in favor of the pro-immunogenic effects (Figure 2).

## 5. In Situ Auto-Vaccination Generated by Radiation

One hallmark of cancer cells is their ability to evade immune surveillance [87]. This evasion is mediated by several different activities, including evading detection and creating an immunosuppressive tumor microenvironment [88]. It is possible that radiation “unmasks” the tumor, thereby making it accessible to both the innate and adaptive immune systems [89].

There are two signals related to the mechanisms of the immune system that are relevant here: Recognition and danger signals. First, radiotherapy induces apoptosis and necrosis in tumor cells, causing them to release tumor antigens, especially neoantigens, into the bloodstream, which may facilitate immune recognition [90]. Tumor antigens can be classified into five categories: (i) Viral antigens, (ii) differentiation antigens, (iii) cancer-germline antigens, (iv) overexpressed antigens, and (v) neoantigens (Figure 3) [91,92,93]. As antigens (i)–(iv) are expressed in normal tissues, these mutations are more likely to induce immunological tolerance and are less likely to elicit effective anti-tumor immune responses [94]. However, neoantigens are expressed in cancer cells owing to genomic mutations altering the protein sequence. This type of antigen is tumor-specific and can elicit an immune response sufficient to clear tumor cells when activated [95,96,97,98].

In addition to releasing the tumor neoantigens, radiation also mediates the translocation of certain endoplasmic reticulum proteins to the cell membrane before apoptosis, including calreticulin [51]. As a danger signal, damage-associated molecular pattern molecules (DAMPs), such as high-mobility group box 1 (HMGB1) and adenosine triphosphate, are released from the cytoplasm of the irradiated tumor cells into the extracellular environment, allowing dendritic cells (DCs) to recognize dying cells and phagocytose them [51,99,100,101,102,103,104,105,106,107]. HMGB1 induces the maturation of DCs, allowing DCs to present antigens more efficiently to T cells [51,101,108]. This process is mediated by type I IFN and works by sensing cancer cell-derived DNA [68,109,110]. The activated DCs migrate to local lymph nodes, and naïve T cells are presented and stimulated with antigens specific to the tumor cells, which results in them becoming effector T cells and returning to the tumor tissue, attracted by chemokines created by the response to the irradiation [61,75].

Thus, the irradiated tumor can become a source of tumor antigens in a process described as in situ auto-vaccination. In addition, the intercellular adhesion molecule-1 (ICAM-1), Fas death receptor, and major histocompatibility complex class I antigen-presenting molecules expressed on irradiated tumor cells allow easy recognition of tumor cells by activated antitumor effector T cells, especially CD8+ T cells [111], which kill them [112]. It has been hypothesized that radiation boosts the immune response by these processes.

Figure 4 summarizes the current understanding of the mechanisms underlying the regression of distant metastatic tumor lesions along with locally irradiated tumors.

Poleszczuk et al. generated a mathematical model to predict the immune-mediated abscopal effects [113]. To construct the model, they hypothesized that abscopal responses can only be achieved if T cells activated at one tumor site approach each of the other metastatic sites in sufficient numbers. In addition, they assumed that trafficking of activated T cells from the site of activation (the irradiated tumor) to a particular organ containing metastatic tumor cells is potentially determined on the basis of physiological blood flow to that organ and by the initial imprinting of T cells by tumor antigen-presenting dendritic cells. Success in triggering the abscopal effect depends, at least in part, on which metastatic site is selected for radiotherapy. Using these parameters, they calculated the “immunogenicity index” of metastatic sites in virtual patients. They proposed that their model can be applied in a patient-specific manner to identify focal therapeutic targets that are most likely to trigger an abscopal response [113].

## 6. Modification of the Abscopal Effect by Immune Checkpoint Inhibitors

Although preclinical studies have involved various ICBs, most of the clinical data regarding the use of radiotherapy with immunotherapy are limited to combinations of radiotherapy and either anti-CTLA-4 or anti-PD-1 agents (Table 1).

CTLA-4 is considered to be one of the major negative immunomodulatory receptors that attenuate T-cell activation [11,140,141,142]. Blocking CTLA-4 enhances T-cell activation, increasing the ratio of CD8+ T cells to regulatory T cells [143] and thereby strengthening the in situ vaccination effect (Figure 1) [142]. In this context, a combination of ipilimumab with radiotherapy is being increasingly valued by researchers and clinicians owing to encouraging results obtained from studies in both mice and humans [6,11].

PD-1 is expressed on the plasma membrane of T cells, DCs and NK cells. PD-L1 and PD-L2, the cognate ligands of PD-1, are expressed in both tumor cells and immune cells. T-cell-mediated signaling pathways are inhibited through the ligation of PD-1 to PD-L1 and PD-L2. PD-1 ligation to PD-L1 primarily inhibits T cell proliferation by blocking cell cycle progression, thus protecting tumor cells from the T-cell attack (Figure 1) [11,14,144,145,146]. A recent study suggested that radiation-induced double-strand breakage of DNA results in the upregulation of the expression of PD-L1 on tumor cells via ATM/ATR/Chk1 kinases [147]. The strong expression of PD-L1 on tumor cells is associated with an improved responses to anti-PD-L1 therapy [148], and radiation may therefore serve as an effective neoadjuvant treatment to increase the effectiveness. Thus, a combination of PD-1 blockade and radiotherapy can theoretically result in a more effective antitumor response, whereas radiotherapy alone elicits the abscopal effect in only a very small proportion of patients [40,149,150].

## 7. Clinical Applications of the Abscopal Effect

Recently, several studies have described elicitation of the abscopal effect by the combined use of radiotherapy and immunotherapy (Table 1). In a study involving patients with melanoma that had metastasized to the brain, Qian et al. reported that concurrent immunotherapy with anti-PD-L1 and anti-CTLA-4 administered within four weeks of stereotactic radiosurgery led to an improved response of the brain lesions compared with treatments at an interval of >4 weeks [151]. In the PACIFIC trial that investigated the effect of the anti-PD-L1 agent durvalumab, better progression-free survival was observed when durvalumab therapy was initiated within 14 days of completing chemoradiotherapy compared than when it was initiated after 14 days [133]. Thus, timing is an important factor in obtaining the abscopal effect. The abscopal effect has mostly been reported in patients who received radiotherapy while receiving concomitant immunotherapy or just before the immunotherapy (Table 1) [11]. The optimal scheduling of radiotherapy and immunotherapy needs to be clearly established, ideally through clinical trials [47].

The dose and fraction of radiotherapy are also key factors that determine the intensity of the abscopal effect. In a meta-analysis, Marconi et al. showed that the abscopal effect is linked to the biologically effective dose of radiation, with a dose of 60 Gy being associated with a 50% probability of achieving the abscopal effect [45,48]. Lugade et al. demonstrated that a 15-Gy single-dose regimen resulted in the production of more tumor-infiltrating T cells than a fractionated regimen [62]. Siva et al. reported that a single dose of radiotherapy (12 Gy) used in combination with immunotherapy did not deplete the established tumors of the effector cells (T and NK cells) that are critical for delivering the effect [27]. In a study of murine lung cancer, a radiation regimen involving five fractions of 10 Gy induced a more robust abscopal effect than that involving 12 fractions of 2 Gy [30]. In preclinical breast cancer models, the abscopal effect was induced only by fractionated radiation and not single-dose radiation when administered in combination with anti-CTLA-4 [22]. A preclinical study of human prostate cancer cells demonstrated that exposure to multifractionated radiation (10 fractions of 1 Gy) induced the release of DAMPs more robustly than single-dose radiation (10 Gy) [152]. Similarly, in a murine melanoma model, Schaue et al. found that fractionated treatment with medium-sized radiation doses of 7.5 Gy/fraction produced the best tumor control and antitumor immune responses [153]. Based on these findings, radiotherapy with ablative high-dose per fractionation has been considered to be a better treatment protocol for enhancing the anti-tumor immune response than conventional fractionation [154]. In this context, many of the clinical trials that have been designed to evaluate the systematic anti-tumor effect of combination immunotherapy and radiotherapy have been designed with hypofractionated radiotherapy (Table 1).

Regarding the benefits of combined radiotherapy and immunotherapy, antitumor effects are potentially synergistic during the combination therapy. This combination is potentially more effective than other conventional therapies and may serve as radical treatment for aggressive or advanced malignant diseases that are otherwise difficult to treat. However, regarding the demerits, grade three or higher toxicity has been reported in several studies. When combining ipilimumab with SABR, 34% of patients experienced grade three or higher treatment-related toxicity [129]. In a phase I study, 11 of 16 patients receiving ipilimumab and radiotherapy experienced grade three toxicities [136]. Radiation-induced necrosis was observed in patients receiving anti-PD-1 therapy for melanoma metastasis to the brain [155]. Furthermore, nivolumab-induced radiation recall pneumonitis was reported in two patients with NSCLC long after thoracic radiotherapy [156]. Among 30 patients treated with radiation and GM-CSF, 6 patients experienced grade three to four fatigue, 10 patients experienced grade three to four hematological adverse effects, and one patient was hospitalized for pulmonary emboli [39]. Since the mechanism underlying abscopal and adverse effects is complex, elucidation of potential methods to decrease the risks is important and urgent, requiring careful examination through clinical trials.

## 8. Future Perspectives

The interplay between radiation and immunotherapy is highly complex, and both the tumor microenvironment and vasculature can be effectively modified by radiotherapy and/or immunotherapy [157]. The pros and cons of radio-immunotherapy can be summarized as follows. Pros: Antitumor effects can be additive or even synergistic on combinatorial administration of these two therapies; cons: (i) Clinical efficacy is usually difficult to predict, and to date, only a small portion of patients experienced some benefits of combinatorial therapy, and (ii) grade three or higher toxicity has been reported. To clinically apply this novel therapy, prospective clinical studies are required to demonstrate induction of the abscopal effect using a combination of radiotherapy and an ICB agent to confirm its clinical impact [150]. The toxicity arising from ICB combined with radiation also needs to be examined, hopefully with a means of prevention or at least amelioration found, considering the observed increase in adverse events from immunotherapy in combination with radiotherapy. Thus, the balance between the benefits and risks needs to be explored carefully. Careful planning regarding the timing, fractionation and doses of radiation to be given with immunotherapy is crucial for establishing an effective utilization of this novel therapeutic strategy.

Furthermore, novel combinations using other immunomodulatory agents or engineered T cells are actively being investigated in preclinical models. OX40 (CD134), a co-stimulatory molecule and member of the tumor necrosis factor receptor superfamily, is expressed on the surface of T cells, where it interacts with OX40L (CD252) expressed on activated antigen-presenting cells [158]. OX40 activation exerts its effects on diverse components of the immune system. OX40 agonistic antibodies increase effector T-cell survival [159,160] and OX40 activation prevents the production of new Tregs and impairs their suppressive effects [161,162]. In preclinical models of lung cancer, Yokouchi et al. reported that the OX40 agonist, in combination with radiotherapy, resulted in an overall survival rate of 80% in 100 days, as opposed to 0% in mice treated with either modality alone [163]. In that study, combinatorial therapy stimulated the recruitment of tumor antigen-specific OX40+ T cells to draining lymph nodes. Similar results were obtained in preclinical combinations of agonistic anti-OX40 with radiotherapy for glioma in C57Bl/6 mice [164]. Moreover, 50–80% of mice administered combinatorial therapy in this model had durable responses and significant survival benefits.

Recently, chimeric antigen receptor (CAR) T-cell therapy has emerged as a potentially curative therapy in treating a broad range of malignant tumors [165]. CAR T-cell therapy has recently gained increasing attention after its success in treating acute and chronic leukemias [166]. Normally, T cell receptors (TCR) must bind to cognate antigens presented in the context of MHC for specific T cells for their activation. T cells engineered to express CARs can directly target a particular antigen without requiring this MHC–TCR interaction, thereby allowing CAR-T cells to eliminate tumor cells more efficiently after encountering antigens [166]. Preclinical studies have justified the combinatorial administration of radiation with CAR-T-cell infusion for cancer therapy [167,168,169]. Thus, the use of these engineered T cells along with radiotherapy is quite promising; however, certain questions remain to be solved through in vivo studies and clinical trials before the implementation of these treatments in the clinical setting.

Furthermore, the synergistic effects of the radiotherapy and immunotherapy may best be realized based on a better understanding of the molecular biology of the abscopal effect. As already described, radiation alone is often insufficient to overcome the existing immunosuppressive tumor microenvironment. Moreover, radiation itself frequently elicits immunosuppressive effects, such as the infiltration of MDSCs and Tregs, which may abrogate its immunostimulatory effect, at least in part. Preclinical and clinical investigation aimed at shifting the balance in favor of proimmunogenic global effect of radiation needs to rather urgently be performed (Figure 2).

Considering the complexity of the responses in the different types of checkpoint blockades and different cancers, developing novel biomarkers for assessing the treatment response, especially during the early course of treatment, is also an urgent requirement for realizing the potential of “radio-immunotherapy” [47,170]. Carefully designed exploratory and definitive studies of potential biomarkers are essential to allow this new treatment strategy to be effectively applied in the clinical setting.

## 9. Conclusions

A combination of radiotherapy with immune checkpoint inhibitors has resulted in a new clinical antitumor strategy that deliberately evokes the abscopal effect. The mechanism underlying this effect may involve enhancement of the immune response by irradiation-induced chemokines and cytokines through the stimulation of antigen-presenting cells and effector cytotoxic T cells, which are amplified by the use of immune checkpoint inhibitors. With the advance in molecular and clinical medicine with regard to this novel approach, the abscopal effect can become a useful paradigm for cancer treatment.

## Figures and Tables

**Figure 1 vaccines-07-00100-f001:**
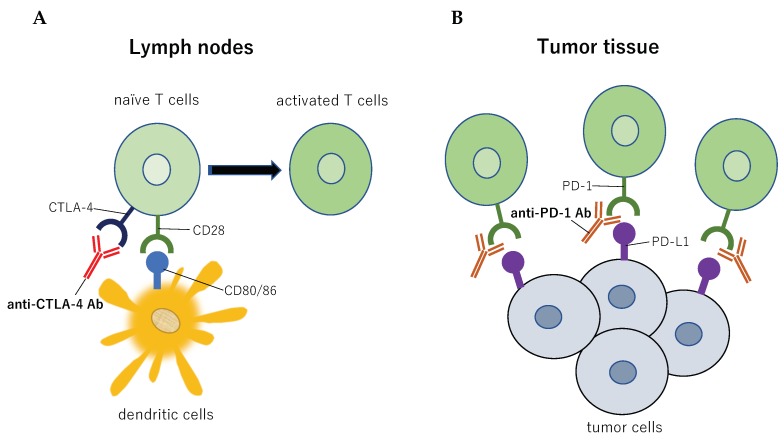
Targets of immune checkpoint inhibitors. (**A**). Sensitization phase. When anti-CTLA-4 antibody binds to CTLA-4, the inhibitory effect on T cells is suppressed. Consequently, effector T cells are induced. (**B**). Effector phase. PD-1 blockade reverses immune evasion mediated by the interaction of PD-1+ immune cells and PD-L1+ tumor cells. CTLA-4, cytotoxic T-lymphocyte-associated protein 4; PD-1, programmed cell death-1; PD-L1, programmed cell death-ligand 1.

**Figure 2 vaccines-07-00100-f002:**
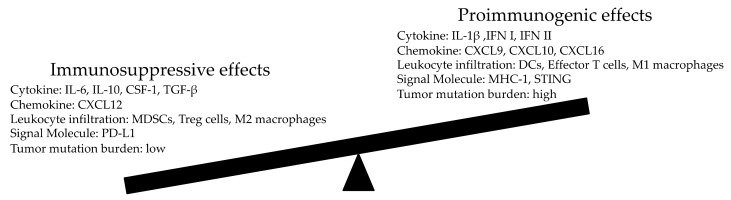
The opposing effects of radiotherapy on the tumor microenvironment. An immunosuppressive microenvironment usually exists in the tumor. Ideally, drug combinations should shift the balance toward the pro-immunogenic and away from the immunosuppressive effect of radiotherapy to favor its pro-immunogenic effects and or abrogate the immune-suppressive ones. RT; radiotherapy, IFN; interferon, IL; interleukin, TGF; transforming growth factor, CSF; colony-stimulating factor, CXCL; CXC-motif chemokine ligand, DCs; dendritic cells, MDSCs; myeloid-derived suppressor cells, Treg; regulatory T lymphocytes, MHC; major histocompatibility complex, STING; stimulator of interferon genes, PD-L1; programmed cell death-ligand 1.

**Figure 3 vaccines-07-00100-f003:**
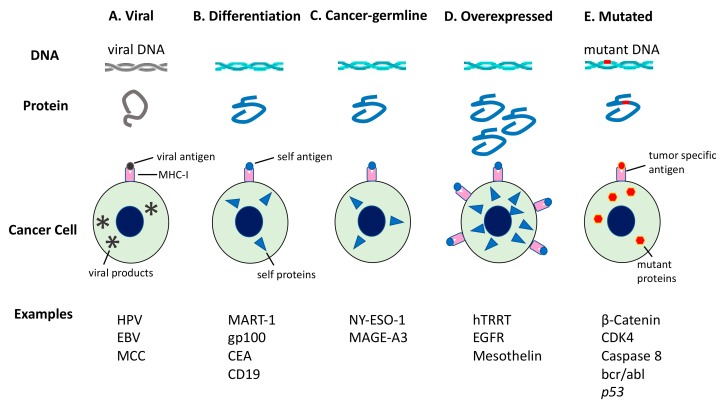
Tumor antigens recognized by immune cells. Tumor antigens are classified in accordance with the pattern of gene expression. The production of antigenic peptides by cancer cells is shown herein. Viral antigens are only expressed in virus-infected cells. Differentiation antigens are encoded by genes with tissue-specific expression. Cancer germline genes are expressed in tumors or germ cells owing to whole-genome demethylation. Some genes are overexpressed in tumors owing to increased transcription or gene amplification. The resulting peptides are upregulated on these tumors but also show a low level of expression in some healthy tissues. However, mutated genes may yield a mutant peptide (neoantigen), which is recognized as non-self by immune cells. CEA, carcinoembryonic Ag; EGFR, epidermal growth factor receptor; EBV, Epstein-Barr virus; HPV, human papillomavirus; hTERT, human telomerase reverse transcriptase; MAGE-A3, melanoma-associated antigen 3; MART-1, melanoma antigen recognized by T cells-1; MCC, Merkel cell carcinoma.

**Figure 4 vaccines-07-00100-f004:**
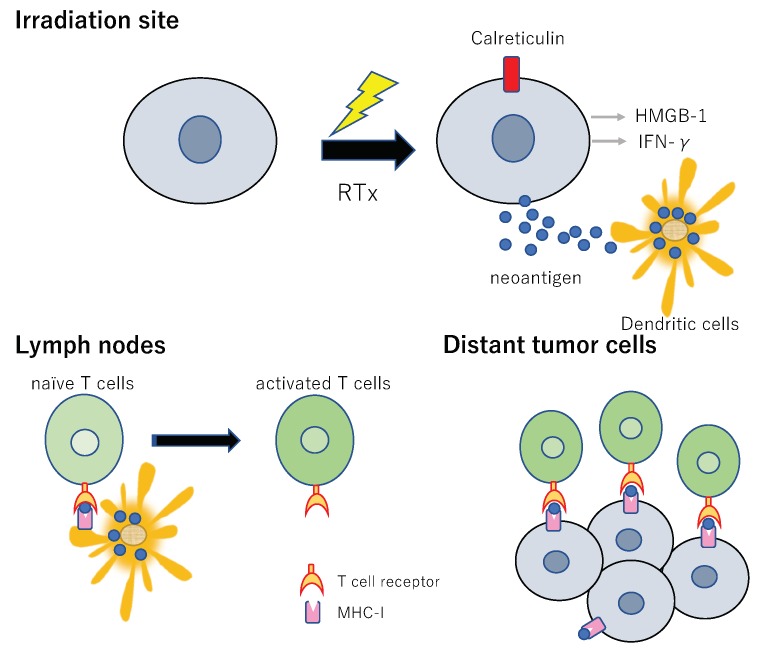
Mechanisms by which radiation enhances the immune response. Cancer-specific peptides released from radiation-damaged cancer cells facilitate the uptake and presentation of antigens by dendritic cells. The radiation enhances the pro-phagocytosis signal (calreticulin) that mediates phagocytosis by macrophages and dendritic cells, thereby increasing the presentation of antigens and the priming of T cells. Following radiation, damaged DNA is released from the nucleus into the cytosol, triggering the cGAS–STING pathway, which activates interferon gene transcription. Due to the stress response, irradiated tumor cells release various mediators, such as IFNγ and HMGB-1. IFNγ upregulates the MHC-I expression on tumor cells and APCs (antigen-presenting cells), including macrophages and dendritic cells. The APCs interact with tumor antigens and then migrate to the lymph nodes, where they present antigens to T cells, a process that is mediated by the MHC pathway and is enhanced by HMGB-1. T cells, especially CD8+ T cells, are activated and begin to propagate. As a result, activated effector T cells exit the lymph nodes and home to the tumors, including both the primary tumors and non-irradiated tumor metastases, to exert their killing effect. The upregulation of MHC-I expression on tumor cells after irradiation facilitates the recognition of the tumor cells by T cells.

**Table 1 vaccines-07-00100-t001:** Selected studies using combinations of radiotherapy and immunotherapy.

Year	Author [ref.]	Study Type	Tumor Type	Number of Patients	IT	RT Type	RT Dose (Gy)	Sequence
2013	Slovin et al. [114]	Phase I/II	Prostate cancer	50	Ipilimumab	SABR	8	RT, IT
2013	Mathew et al. [115]	Retrospective	Melanoma	25	Ipilimumab	SRS	20	various
2013	Barker et al. [116]	Retrospective	Melanoma	29	Ipilimumab	SABR	24–62.5	Concurrent
2013	Silk et al. [117]	Retrospective	Melanoma	17	Ipilimumab	SRS/WBRT	14–37.5	RT, IT or IT, RT
2014	Kwon et al. [118]	phase III	Prostate cancer	799	Ipilimumab	SABR	8	RT, IT
2015	Twyman-Saint Victor et al. [119]	phase I	Melanoma	22	Ipilimumab	SABR	12–24	RT, IT
2015	Kiess et al. [120]	Retrospective	Melanoma	46	Ipilimumab	SRS	15–24	various
2015	Patel et al. [121]	Retrospective	Melanoma	54	Ipilimumab	IMRS	15–21	various
2015	Tazi et al. [122]	Retrospective	Melanoma	10	Ipilimumab	SRS	various	various
2016	Hiniker et al. [123]	phase I	Melanoma	22	Ipilimumab	SABR/IMRT/3D	18–50	Concurrent
2016	Qin et al. [124]	Retrospective	Melanoma	44	Ipilimumab	SABR/CEBRT	18–42	RT, IT or IT, RT
2016	Ahmed et al. [125]	Retrospective	Melanoma	96	Ipilimumab/Nivolumab/Pembrolizumab	SRS	15–24	various
2016	Ahmed et al. [126]	Retrospective	Melanoma	26	Nivolumab	SRS	16–30	various
2016	Levy et al. [127]	Retrospective	various	10	Durvalumab	SRS/3D	6–92	Concurrent
2016	Qian et al. [128]	Retrospective	Melanoma	75	Ipilimumab, Pembrolizumab	SRS	12–24	various
2017	Tang et al. [129]	phase I	various	35	Ipilimumab	SABR	50–60	Concurrent
2017	Skrepnik et al. [130]	Retrospective	Melanoma	25	Ipilimumab	SRS	16–24	various
2017	Koller et al. [131]	Retrospective	Melanoma	70	Ipilimumab	CEBRT	48	Concurrent
2017	Bang et al. [132]	Retrospective	Melanoma, Lung cancer, Renal cell carcinoma	133	Ipilimumab/Nivolumab/Pembrolizumab	SABR/SRS/ IMRT/WBRT	8–66	various
2017	Antonia et al. [133]	phase III	Lung cancer	713	Durvalumab	CRT	54–66	RT, IT
2017	Shaverdian et al. [46]	phase I	Lung cancer	98	Pembrolizumab	various	various	RT, IT
2017	Aboundaram et al. [134]	Retrospective	Melanoma	17	Pembrolizumab	various	24–45	Concurrent
2017	Anderson et al. [135]	Retrospective	Melanoma	21	Pembrolizumab	SRS/WBRT	18–21	Concurrent
2017	Williams et al. [136]	phase I	Melanoma	16	Ipilimumab	SRS/WBRT	15–30	Concurrent
2018	Roger et al. [137]	phase I	Melanoma	25	Nivolumab/Pembrolizumab	SABR/SRS	26	Concurrent or RT, IT
2018	Martin et al. [138]	Retrospective	various	115	Ipilimumab/Nivolumab/Pembrolizumab	SABR/SRS	18–30	various
2018	Luke et al. [139]	phase I	various	79	Pembrolizumab	SABR	10–15	RT, IT

IT, immunotherapy; RT, radiotherapy; ref., reference; SRS, stereotactic radiosurgery; SABR, stereotactic ablative body radiotherapy; WBRT, whole-brain radiotherapy; IMRS, intensity-modulated radiosurgery; IMRT, intensity-modulated radiotherapy; 3D, three-dimensional conformal-radiation therapy; CEBRT, conventional external beam radiation therapy; CRT, chemoradiotherapy; n.d., not described; PD-1, programmed cell death-1; CTLA-4, cytotoxic T-lymphocyte-associated protein 4.

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
