# Peer review of "Radiation as an In Situ Auto-Vaccination: Current Perspectives and Challenges"

_vaccines, 2019, doi:10.3390/vaccines7030100_

Round 1
Reviewer 1 Report
The author reviewed the interplay between cancer immunotherapy and radiotherapy. The review is interesting, but needs some improvements:
- The authors should discuss the interplay between cancer immunotherapy and radiotherapy also on a pathway level relevant for the central cancer immune, apoptosis and proloferation pathways, especially in the light of mutational differences between different tumor entities influencing pathway activation.
- Discuss in more detail the current method approaches for designing novel combination therapies for precision medicine (in vitro, in silico) as it is known that results from pre-clinical research cannot always be translated to the clinic. Also regarding the toxicity.
- Discuss in more detail the pros and cons of radio-immunotherapy compared to other treatments (e.g. with a table)
Author Response
The author reviewed the interplay between cancer immunotherapy and radiotherapy. The review is interesting, but needs some improvements:
- The authors should discuss the interplay between cancer immunotherapy and radiotherapy also on a pathway level relevant for the central cancer immune, apoptosis and proloferation pathways, especially in the light of mutational differences between different tumor entities influencing pathway activation.
Response: We agree with your comment. Accordingly, we included an explanation regarding the interplay between cancer immunotherapy and radiotherapy with respect to the p53mutation and pathways involved in cancer progression.
- Discuss in more detail the current method approaches for designing novel combination therapies for precision medicine (in vitro, in silico)as it is known that results from pre-clinical research cannot always be translated to the clinic. Also regarding the toxicity.
Response: Regarding the in silico analysis performed herein to design novel combination therapies for precision medicine,we additionally described the mathematical model to predict the abscopal effect previously advocated by Poleszczuk et al.
Furthermore, regarding the in vitroanalysis, we described the combination of radiotherapy with a new drug OX40 and CAR-T cell-based therapy, using engineered T cells, in the future perspectives section.
- Discuss in more detail the pros and cons of radio-immunotherapy compared to other treatments (e.g. with a table)
Response: We described the pros and cons of radio-immunotherapy in the clinical applications of the abscopal effect section and summarized them in the future perspectives section.
Thank you very much for your valuable comments.
Reviewer 2 Report
This review on the abscopal effect of radiotherapy potentially provides a useful resource on instances where this had been documented, most recently and most commonly when irradiation is followed by immunomodulatory antibody therapy with anti-CTLA-4 or anti-PD-1/PD-L1. A superficial description of the basic biology and consensus opinion on mechanisms is provided, but no new concepts are introduced and none of the author´s own work is described. The very long reference list does not always reflect the optimum citation for the point the author is making, indicating less than perfect understanding of the field. General statements are not always helpful or completely accurate.
Specific details:
The abstract states that “To date, there is no consensus regarding how tumor regression is obtained via the abscopal effect” but there is indeed a consensus (which the author goes on to discuss in the paper). Where is the controversy here?
Introduction – please cite Mole 1953
In the Intro, place the paragraph on CTLA-4 before PD-1, which has more or less displaced it nowadays. It is not the case that CTLA-4 is just another “Another key component in ICB…” – it was the first and now only used in combination with anti-PD-1
Fig. 1 is good. The crucial thing is what influences the balance? Please provide some indications of how investigators in this area view the control of this balance. What is your own contribution?
Line 170 states “These DNA alterations generated by radiation may act as a rich source of neoantigens that increase immune surveillance”. This sounds as if you mean that the mutagenic affects of radiation cause neoantigen generation, but that is not the case, is it? Please clarify.
Lines 117 and later 180 both refer to cGAS and STING – this is repetitive
Fig. 2: there is a reproduction problem, please check
Line 200: HMGB-1 is not a cytokine
Line 223: you state that the combination of ipi and radiotherapy has FDA approval – please elucidate
Line 227: “PD-1 ligation to PD-L1 mainly promotes T-cell apoptosis”. This is incorrect – the ref. you cite (Ref. 6) explicitly states that is does not.
Author Response
The abstract states that "To date, there is no consensus regarding how tumor regression is obtained via the abscopal effect" but there is indeed a consensus (which the author goes on to discuss in the paper). Where is the controversy here?
Response: We agree with your comment. We deleted and revised some sentences to prevent confusion among the readers.
Introduction – please cite Mole 1953
Response: We cited Mole’s study in the text.
In the Intro, place the paragraph on CTLA-4 before PD-1, which has more or less displaced it nowadays. It is not the case that CTLA-4 is just another "Another key component in ICB..." – it was the first and now only used in combination with anti-PD-1
Response: We changed the order, such that the explanation regarding CTLA-4 is present before that of PD-1, per your suggestion.
Fig. 1 is good. The crucial thing is what influences the balance? Please provide some indications of how investigators in this area view the control of this balance. What is your own contribution?
Response: Numerous studies have indicated that radiotherapy alone is unfortunately seldom successful in shifting the balance towards immune activation, as reported through rare cases of abscopal effects in the clinical setting. We included these explanations.
We have been working on the sequencing of the genome of lung cancer and reported some findings regarding the tumor mutation burden and its effect on clinical outcomes. In our revised manuscript, we further stated that the abscopal effect can be expected in cancers with a high tumor mutation burden, and we cited our previous studies.
Line 170 states "These DNA alterations generated by radiation may act as a rich source of neoantigens that increase immune surveillance". This sounds as if you mean that the mutagenic affects of radiation cause neoantigen generation, but that is not the case, is it? Please clarify.
Response: Thank you for your pragmatic suggestion. We revised these sentences to not include the mutagenic effects of radiation.
Lines 117 and later 180 both refer to cGAS and STING – this is repetitive
Response: We deleted the repeated sentence.
Fig. 2: there is a reproduction problem, please check
Response: We replaced one figure, and all illustrations are now our own constructs.
Line 200: HMGB-1 is not a cytokine
Response: We changed the word "cytokines" to "mediators."
Line 223: you state that the combination of ipi and radiotherapy has FDA approval – please elucidate
Response: We confirmed that no FDA approval is available for combinatorial immunotherapy and radiotherapy. We apologize for our fundamental misunderstanding. We deleted this sentence.
Line 227: "PD-1 ligation to PD-L1 mainly promotes T-cell apoptosis". This is incorrect – the ref. you cite (Ref. 6) explicitly states that is does not.
Response: We agree with your comment that PD-1 ligation to PD-L1 does not induce T-cell apoptosis; however, it primarily inhibits T-cell proliferation. Accordingly, we replaced the sentence with an appropriate description.
Thank you again for your constructive comments.
Reviewer 3 Report
Dear Author,
In the review article from Taichiro Goto entitled ‘Radiation as an in situ auto-vaccination: Current Perspectives and Challenges.’, the author summarized the abscopal effect of radiation therapy and immunotherapy. Overall, the review article is well organized and information is summarized in an sophisticated fashion; however, there are several critical points should be improved. Specific points are listed as followings.
Major points are:
1) Figure 1. The figure is OK, but several factors including cytokines and cells and cell surface molecules and signaling molecules are just listed. I am afraid this presentation may confuse readers. I would like to recommend that factors should be presented in sectioned fashion such as: Immunosuppressive cytokines: IL-6, TGF-beta so on…
2) Page 5-6, line 167-171. Only radiation induced DNA damage does not release neoantigens. Cell death by apoptosis or necrosis is necessary. The description cause confusion for authors. Radiation may increase neoantigen due to the DNA damage. This is aggregable. However, the effect is just on the radiation target lesion and this is not a mechanism to account abscopal effect.
3) Figure 2. The author showed RTx increase neoantigen release. But, neoantigen is not only tumor-associated antigens (TAAs). There are several other types of TAAs including cancer-testis antigens, over-expressed antigens and differentiation antigens. Other types of antigens should also be described in the figure and in the text.
4) PD-1 and CTLA-4, targets of immune checkpoint inhibitors should be shown and explained in another figure to facilitate readers understanding.
5) Page 9. Line 227. PD-L1 and PD-L2 are expressed in also immune cells, should be described.
Yours sincerely,
Author Response
Reviewer 3:
Figure 1. The figure is OK, but several factors including cytokines and cells and cell surface molecules and signaling molecules are just listed. I am afraid this presentation may confuse readers. I would like to recommend that factors should be presented in sectioned fashion such as: Immunosuppressive cytokines: IL-6, TGF-beta so on…Response: We modified the figure per your suggestion.
Page 5-6, line 167-171. Only radiation induced DNA damage does not release neoantigens. Cell death by apoptosis or necrosis is necessary. The description cause confusion for authors. Radiation may increase neoantigen due to the DNA damage. This is aggregable. However, the effect is just on the radiation target lesion and this is not a mechanism to account abscopal effect.Response: We agree that this sentence may confound the readers. We revised these sentences to not include the mutagenic effects of radiation. Moreover, tumor suppression due to the abscopal effect is probably regulated by a systemic antitumor effect caused by the release of cytokines and neoantigens into the bloodstream. We included these explanations in the text.
Figure 2. The author showed RTx increase neoantigen release. But, neoantigen is not only tumor-associated antigens (TAAs). There are several other types of TAAs including cancer-testis antigens, over-expressed antigens and differentiation antigens. Other types of antigens should also be described in the figure and in the text.Response: We included some explanations in the text and a figure to explain tumor antigens.
PD-1 and CTLA-4, targets of immune checkpoint inhibitors should be shown and explained in another figure to facilitate readers understanding.Response: We included some descriptions in the text and a figure to explain the mechanism of action of immune checkpoint inhibitors.
Page 9. Line 227. PD-L1 and PD-L2 are expressed in also immune cells, should be described.Response: We agree with your comment. Accordingly, we revised this phrase.
Thank you again for your helpful comments.
Round 2
Reviewer 3 Report
Dear Author,
The author addressed concerns.
Sincerely yours,